# Retrieval of surface solar irradiance from satellite imagery using machine learning: pitfalls and perspectives

**Hadrien Verbois, Yves-Marie Saint-Drenan, Vadim Becquet, Benoit Gschwind, and Philippe Blanc**

Mines Paris, Université PSL, Centre Observation Impacts Energie (O.I.E.), 06904 Sophia Antipolis, France

**Correspondence:** Hadrien Verbois (hadrien.verbois@minesparis.psl.eu)

**Abstract.** Knowledge of the spatial and temporal characteristics of solar surface irradiance (SSI) is critical in many domains. While meteorological ground stations can provide accurate measurements of SSI locally, they are sparsely distributed worldwide. SSI estimations derived from satellite imagery are thus crucial to gain a finer understanding of the solar resource. Inferring SSI from satellite images is, however, not straightforward, and it has been the focus of many researchers in the past 30 to 40 years. For long, the emphasis has been on models grounded in physical laws with, in some cases, simple statistical parametrizations. Recently, new satellite SSI retrieval methods have been emerging, which directly infer the SSI from the satellite images using machine learning. Although only a few such works have been published, their practical efficiency has already been questioned.

The objective of this paper is to better understand the potential and the pitfalls of this new family of methods. To do so, simple multi-layer-perceptron (MLP) models are constructed with different training datasets of satellite-based radiance measurements from Meteosat Second Generation (MSG) with collocated SSI ground measurements from Météo-France. The performance of the models is evaluated on a test dataset independent from the training set in both space and time and compared to that of a state-of-the-art physical retrieval model from the Copernicus Atmosphere Monitoring Service (CAMS).

We found that the data-driven model's performance is very dependent on the training set. Provided the training set is sufficiently large and similar enough to the test set, even a simple MLP has a root mean square error (RMSE) that is 19 % lower than CAMS and outperforms the physical retrieval model at 96 % of the test stations. On the other hand,

in certain configurations, the data-driven model can dramatically underperform even in stations located close to the training set: when geographical separation was enforced between the training and test set, the MLP-based model exhibited an RMSE that was 50 % to 100 % higher than that of CAMS in several locations.

## 1 Introduction

Spatial and temporal variabilities in solar surface irradiance (SSI) are of great interest across a range of fields, including climatology, solar energy, health, architecture, agriculture, and forestry. SSI estimations can be made using solar radiation measurements from existing networks of meteorological ground stations. However, these are sparsely distributed worldwide. Spaceborne imaging systems of meteorological geostationary satellites represent complementary upwelling radiance sources for SSI retrieval, as they enable better spatial and temporal coverage (Blanc et al., 2017; Müller and Pfeifroth, 2022; Tournadre, 2020). Since the 1980s, multiple SSI retrieval approaches have been proposed using satellite images, from the earlier cloud index methods (Cano et al., 1986; Rigollier and Wald, 1998) to more recent approaches relying on advanced radiative transfer models (Xie et al., 2016; Qu et al., 2017). Some of these retrieval algorithms are operational and provide SSI estimations worldwide. For example, HelioClim3 (Blanc et al., 2011a) offers real-time estimations of the global horizontal irradiance (GHI) over Africa and Europe. CAMS, the Copernicus Atmosphere Monitoring Service, is another near-real-time service that derives SSI estimations from data collected by both Meteosat and Himawari satellites; it covers areas

including Africa, Europe, and a significant portion of Asia (Schroedter-Homscheidt et al., 2016). In the United States, the National Solar Radiation Database (NSRDB; Sengupta et al., 2018) serves as a valuable resource, providing SSI estimates primarily from the GOES satellites. The performances of these solar radiation databases vary with the location and sky conditions; they are discussed in detail in Forstinger et al. (2023). Statistical methods have also been developed to post-process these retrieval models and correct their errors based on historical ground measurements (Polo et al., 2016, 2020; Huang et al., 2019). These correction algorithms, however, are mostly based on simple statistical methods and do not aim to replace the physical retrieval models upstream. In addition, most of the correction models proposed in the literature are local and therefore cannot CE1 generalize to locations they have not seen during training (Verbois et al., 2022).

In the past decade, Earth science has been revolutionized by the advent of machine and deep learning (Reichstein et al., 2019; Boukabara et al., 2019), with important developments in remote sensing (Ball et al., 2017), severe weather predictions (McGovern et al., 2017; Racah et al., 2017), and numerical weather modeling (Brenowitz and Bretherton, 2018; Rasp et al., 2018). In the field of SSI retrieval, new data-driven approaches are emerging based on automatic statistical learning, which attempt to infer a direct relationship between satellite images and SSI ground measurements. Papers presenting such retrieval methods report promising performance (Jiang et al., 2019; Hao et al., 2019, 2020). However, a more thorough analysis is needed. In particular, the ability of machine-learning-based models to generalize to new locations and specific meteorological and atmospheric events must be rigorously evaluated. Indeed, when Yang et al. (2022) evaluated the method proposed by Hao et al. (2019) outside the algorithm's training locations, they found that the method was performing significantly worse than expected.

In this work, we propose to further explore the potential of machine-learning-based satellite-retrieval methods and to identify some of the main pitfalls that come with this type of approach. Our objective is not to introduce a new retrieval method; hence, we have deliberately opted for a simple, fully connected architecture. This choice allows our conclusions regarding generalization to extend more effectively to the realm of complex networks (convolutional, recurrent, attention-based, generative, etc.), which are generally prone to encountering greater generalization challenges (Wang et al., 2017; Ranalli and Zech, 2023). We conduct a thorough and critical analysis of its performance and compare it with a state-of-the-art retrieval model, Heliosat-4 (Qu et al., 2017), operational as part of the Copernicus Atmosphere Monitoring Service (CAMS) radiation service.

The paper is organized as follows. In Sect. 2, we present the data used in this study. In Sect. 3, we describe our proposed machine-learning-based model. In Sect. 4, we set the stage for our analysis and present the experimental setups.

The results are discussed in Sect. 5. Discussion and conclusions are given in Sect. 6.

## 2  Data

In this section, we briefly describe the data used in this study. Table 1 gives an overview.

### 2.1  Satellite observation

We have been using readings of upwelling radiances $L_\lambda$ from the multispectral optical imaging system aboard the Meteosat Second Generation (MSG) meteorological geostationary satellite. MSG has 12 different channels, but we only use 3 of them here: 2 visible bands (centered on 0.6 and 0.8 µm) and 1 infrared band (centered on 10.8 µm). MSG channels have a temporal resolution of 15 min and a spatial resolution of 3 km at nadir $(0, 0)$[1], which above France corresponds to pixels of ca. 4 by 6 km (in the E–W and N–S directions, respectively) (EUMETSAT, 2017).

### 2.2  Ground measurements

#### 2.2.1  Météo-France stations

This study relies for training, validating, and testing on ground SSI measurements from 231 meteorological stations operated by Météo-France and spread over metropolitan France, as shown in Fig. 2. The stations are equipped with Kipp & Zonen thermopile pyranometers[2] that measure 1 min solar surface irradiance (SSI); here, however, we only have access to hourly averages. The data span 9 years, between 2010 and 2019, but not all stations were operational during the whole period.

Strict quality checks (QCs) are applied to the broadband data, as described extensively in Verbois et al. (2023) and summarized in Appendix A. The idea is to select among all the ground measurements of SSI the ones that are the least questionable with both commonly used automatic quality check procedures and expert visually based scrupulous inspection, station by station, day by day.

#### 2.2.2  Carpentras station

The Météo-France station of Carpentras, included in the dataset described in Sect. 2.2.1, is also part of the Baseline Surface Radiation Network (BSRN; Ohmura et al., 1998) and the Aerosol Robotic Network (AERONET; Holben et al., 1998). As a BSRN station, it provides measurements of 1 min

---

[1]Except for the high-resolution visible (HRV) channel, which provides measurements with a resolution of 1 km, but on a reduced portion of the disk. This channel is not used in this study.

[2]The details of the instrument at each station can be found at https://donneespubliques.meteofrance.fr/?fond=contenu& id_contenu=37 (last access: 5 September 2023).

**Table 1.** Overview of data used in this study. TS1

| Data | Time sampling | Spatial resolution | Time extent | Spatial extent | Source |
|------|------|------|------|------|------|
| SSI measurements | 1 h | Punctual | 2010–2019 | 231 locations in France | Courtesy of Météo-France |
| SSI measurements | 1 min | Punctual | 2018–2019 | Carpentras (FR) | BSRN |
| SSI estimations | 1 h | ca. $4 \times 6$ km | 2018–2019 | France | CAMS radiation services |
| CSI estimations | 1 min | ca. $4 \times 5$ km | 2018–2019 | France | CAMS radiation services |
| Climatic albedo | 1 min | ca. $4 \times 5$ km | 2010–2019 | France | CAMS radiation services |
| AOD measurements | 1 min | Punctual | 2018–2019 | Carpentras (FR) | AERONET |

SSI. As an AERONET station, it provides measurements of spectral aerosol optical depth (AOD). The AODs at different wavelengths are measured with a sun photometer but are only valid under clear-sky conditions. Cloud screening is thus applied to the raw data, and measurements are therefore only available intermittently (Giles et al., 2019). In this work, we use the AOD at 500 nm.

## 2.3 Copernicus Atmosphere Monitoring Services

The Copernicus Atmosphere Monitoring Service (CAMS) provides time series for various atmospheric and meteorological variables.

CAMS radiation service provides time series of global, direct, and diffuse ground irradiances. It relies on Heliosat-4, a state-of-the-art physical retrieval method (Qu et al., 2017), to infer ground irradiance from MSG satellites and CAMS atmospheric composition. CAMS estimations of SSI are used as a benchmark in this study. CAMS SSI natively comes with a resolution of 15 min, as it is derived from MSG. Here, however, we use hourly averages of SSI to match the resolution of the ground data we use as a reference (Sect. 2.2).

It should be noted that other physical retrieval methods might outperform CAMS (Forstinger et al., 2023). It remains, nonetheless, a state-of-the-art retrieval model.

CAMS also implements the McClear clear-sky model, which provides estimations of global, diffuse, and direct clear-sky irradiances. It is based on look-up tables from the radiative transfer model libRadtran and fed by partial aerosol optical depth, ozone, and water vapor data from CAMS atmosphere services (Lefèvre et al., 2013; Gschwind et al., 2019). In this work, we use its global component, abbreviated CSI for clear-sky irradiance. As it will be used to detect clear-sky instances at the Carpentras station, we use 1 min values.

We also use the CSI hourly mean to compute clear-sky index $k_c$ from the SSI: $k_c = SSI/CSI$.

## 2.4 Ground albedo

The ground albedo is the fraction of the total irradiance reaching the surface of the Earth that is reflected by the ground. In this work, we use the ground albedo to analyze the performance of the retrieval models. We rely on values derived from MODIS datasets (Blanc et al., 2014).

## 3 Machine-learning-based retrieval model

In this section, we present our proposed machine-learning-based SSI satellite retrieval model – ML model in short. We describe the target and predictors in Sect. 3.1 and 3.2, respectively; the neural network architecture is detailed in Sect. 3.3. Finally, we shortly discuss running time in Sect. 3.4. A code snippet showing the exact implementation of the network in TensorFlow is provided in the Supplement.

## 3.1 Target

The target of the model is the hourly solar surface irradiance (SSI) and more precisely the global horizontal irradiance (GHI) component, which is the downwelling shortwave surface flux. The ground truth, used for training the model and evaluating its performance, is provided by the Météo-France measurement stations described in Sect. 2.2. To accelerate the training, the values are normalized by the corresponding average irradiance over the training period. The inverse transformation (also with the average irradiance over the *training* period) is applied to the network predictions before starting to analyze its performance.

## 3.2 Predictors

The choice of predictors is critical in statistical learning. Because we use a simple fully connected network (Sect. 3.3), we want to keep the dimensions of the predictor set relatively low while giving as much context as possible to the algorithm. We are also restricted by the fact that the ML model must be fully real-time and can therefore only utilize past and present data. To estimate the SSI in each location (with latitude $x$ and latitude $y$) at a given time $t$, predictors from four sources are used.

The main inputs to the ML model come from satellite measurements. We use the upwelling radiances $L_{0.6\,\mu m}$, $L_{0.8\,\mu m}$,

and $L_{10.8\,\mu m}$, described in Sect. 2.1[3]. To give the model as much spatial and temporal context as possible, 9 neighboring pixels and 13 preceding 15 min time steps are used as input. This corresponds to a zone with an area of ca. 12 by 18 km and a period of 3 h. In summary, for a point with latitude and longitude $(x, y)$ at time $t_0$, such that the closest satellite pixel has coordinates $(i_0, j_0)$, the following predictors are taken from MSG data:

$$L_\lambda(i, j, t) \text{ for } i, j \in [i_0 - 1, i_0 + 1] \times [j_0 - 1, j_0 + 1]$$
$$t \in [t_0 - 12\mathrm{d}t, t_0], \text{ where } \mathrm{d}t = 15\,\mathrm{min}$$
$$\lambda \in \{0.6, 0.8, 10.8\,\mu m\}.$$

The solar azimuth angle $\psi_s$ and the solar elevation $\gamma_s$, computed using the sg2 python library (Blanc and Wald, 2012), are also provided as predictors. They define the topocentric angular position of the sun. The day of the year and the hour of the day are given as predictors too. Finally, the latitude and longitude, as well as the corresponding altitude, are used as predictors.

In total, the model has 358 predictors, summarized in Table 2. Each predictor is normalized and centered. These 358 predictors are concatenated in a single 1D vector, which is used as input to the neural network.

### 3.3 The machine learning model: a fully connected network

As discussed in the Introduction, the aim of this work is not to propose a new optimized retrieval model but to investigate the advantage and drawbacks of purely ML-based models. We therefore implement a *classic* algorithm: a fully connected neural network (FCN), or multi-layer perceptron (MLP). This model has been around for many years and has proven very powerful in many fields and industries. It is not, however, the state of the art in machine learning: *deep* architectures optimized for images or time series, for example, have since been developed and outperform FCN for complex spatio-temporal problems. As we see in this paper, a simple FCN is nonetheless sufficient to at least partially solve the satellite retrieval challenge.

Our FCN has the following configuration:

- There is one hidden layer of 64 neurons, for a total of 23 041 parameters.

- The hidden layer uses a rectified linear unit (ReLU) activation function, and the last neuron uses a linear activation function.

- The weights are initialized randomly using a normal distribution.

- The loss function is the mean square error (mse).

The same configuration, but with two and three hidden layers, was also tested. As they had similar (validation) performance, we preferred the simpler configuration.

The network is trained using the RMSprop algorithm with learning rate = 0.001, $\rho = 0.9$, momentum = 0.0, and $\epsilon = 1 \times 10^{-7}$ (Tieleman et al., 2012)[TS2]. Regularization is implemented through an early-stopping procedure, which stops training if the validation error does not decrease for more than 20 epochs.

Because the last layer uses a linear activation function, there is no guarantee that the predicted value is positive. To ensure we do not get any negative SSI estimation, any negative prediction is set to 0.

The random initialization of the network weights slightly impacts the network performance. The impact on the model performance is, however, limited, as discussed in Appendix B. In this study, each model was trained 20 times, with different (randomly assigned) initial weights, and the results for the worst-performing model (in terms of test mse) are presented in the rest of the paper. Choosing the best-performing one – or any of the 20 runs – would lead to very similar results and the same conclusions.

### 3.4 Running time

An important aspect of real-time satellite retrieval methods is their running time. At minimum, the model should not take longer to run on a full image than the satellite update time; for MSG it is 15 min, but for third-generation satellites such as Meteosat Third Generation[CE2], it goes as low as 5 min. For some applications, such as nowcasting and short-term forecasting, estimations are needed as soon as possible, and a processing time way below the satellite update time is beneficial.

Machine learning algorithms, including neural networks, may take a long time to train but usually have short running times. Using a single core and an NVIDIA Tesla A100 80GB, training the ML models presented in this work takes a few minutes (depending on the size of the training set). Applying the models to the full MSG disk ($3712 \times 3712$ pixels) requires less than 2 s on the same machine[4]. As a comparison, CAMS, whose running time varies with the time of day, takes up to 6 min and 30 s on a single core to treat the same inputs.

It should also be noted that while adding extra predictors – for example, more channels or a larger pixel neighborhood – could significantly increase the training time of the ML model, it is likely to only marginally increase its running time.

---

[3]These are the channels mainly used by Heliosat-2 and Heliosat-4. Other wavelengths may nonetheless be useful to a machine-learning-based model, and their impact on model performance should be explored in future works.

[4]This does not include data pre-processing.

**Table 2.** Predictors used to estimate SSI at a given time and place.

| Name | Dimension | Source |
|---|---|---|
| Satellite $L_{0.6\,\mu m}$ | $3 \times 3 \times 13 = 117$ | MSG |
| Satellite $L_{0.8\,\mu m}$ | $3 \times 3 \times 13 = 117$ | MSG |
| Satellite $L_{10.8\,\mu m}$ | $3 \times 3 \times 13 = 117$ | MSG |
| Solar position: elevation and azimuth angles | 2 | sg2 (Blanc and Wald, 2012) |
| Hour of the day and day of the year | 2 | Calendar |
| Location and altitude | 3 | BSRN |

## 4 Experiments' setup

In this section, we describe the setup of the experiments conducted in this study. In Sect. 4.1, we introduce the metrics used to assess the performances of the SSI estimations; in Sect. 4.2, we discuss the splitting of the data into training and test sets; finally, in Sect. 4.3, we describe the clear-sky detection method used in Sect. 5.2.

### 4.1 Performance metrics

The SSI estimations produced by the ML model, $\hat{x}_{ML}$, as well as the SSI estimation from CAMS, $\hat{x}_{CAMS}$, are compared with the ground measurements $x$ of SSI from Météo-France stations. Both datasets have a resolution of 1 h.

Three different error metrics are used, namely the root mean square error (RMSE), the mean bias error (MBE), and the standard deviation of the error (SDE):

$$\text{RMSE} = \sqrt{\frac{1}{n}\sum_{k=1}^{n}\left(\hat{x}_k - x_k\right)^2} \tag{1}$$

$$\text{MBE} = \frac{1}{n}\sum_{k=1}^{n}\left(\hat{x}_k - x_k\right) \tag{2}$$

$$\text{SDE} = \sqrt{\frac{1}{n}\sum_{k=1}^{n}\left(\hat{x}_k - x_k - \text{MBE}\right)^2}, \tag{3}$$

where $n$ is the number of points, and $\hat{x}_k \in \{\hat{x}_k^{ML}, \hat{x}_k^{CAMS}\}$. MBE measures the accuracy – or bias – of the estimations, SDE measures their precision, and RMSE is a combination of both. The three metrics are related as follows: $\text{RMSE}^2 = \text{MBE}^2 + \text{SDE}^2$.

The correlation between $\hat{x}$ and $x$ is also a popular metric. To compare estimations to measurements and quantify the performance of a model, we use Pearson's correlation coefficient $\rho_{pearson}$. Because it measures linear correlation, the Pearson correlation is not appropriate to unveil non-linear relationships between two time series. To quantify the strength and direction of association between two time series, we thus prefer Spearman's rank-order correlation, $\rho_{spearman}$ (Spearman, 1987).

To compare the performance of the ML model and CAMS, we sometimes use the RMSE skill score, taking CAMS as a reference:

$$\text{Skill} = 1 - \frac{\text{RMSE}_{ML}}{\text{RMSE}_{CAMS}}. \tag{4}$$

A positive skill means that $\text{RMSE}_{ML} < \text{RMSE}_{CAMS}$, i.e., that the ML model outperforms CAMS in terms of RMSE.

### 4.2 Training, validation, and test set

Splitting the data into a test and training set is a crucial step in machine learning studies. Machine learning models, such as neural networks, can achieve exceptional performance with data that are similar to the data used for their training. However, their performance may deteriorate drastically when they operate outside their training space (Hastie et al., 2009). The model's ability to generalize to new, unseen data is a crucial metric of its performance. The definition of what constitutes data outside the training space depends on the specific problem at hand, as it varies based on how the model will be used in practice. The training and test set must therefore be selected carefully to ensure the model's suitability for deployment in practical applications.

In this study, we evaluate a satellite retrieval model which is meant to provide accurate SSI estimations in any location – at least within a certain region – and at any (future) time. We must thus ensure that the ML model generalizes in time and space. To that end, we use different locations for training and testing and reserve the period 1 July 2018 to 30 June 2019 for testing, while only data from 1 January 2010 to 30 June 2018 are used for training. The setup, adapted from Verbois et al. (2022), is illustrated in Fig. 1.

How we assign measurement stations to one set or the other is also important and will test the ability of the model to generalize in space differently. In this study, we test four training setups, with different objectives in mind:

- Training setup 1, described in Sect. 4.2.1, allows us to evaluate the ability of the model to generalize in space when training and test stations are geographically intertwined.

- Training setup 2, described in Sect. 4.2.2, allows us to understand the sensitivity of the model performance to the number of training years.

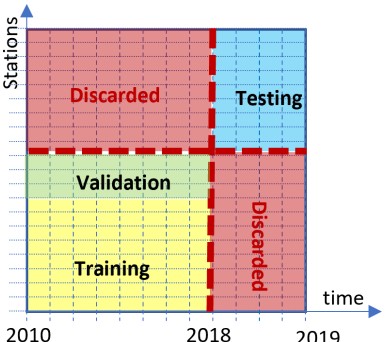

**Figure 1.** Training, validation, and test sets, after Verbois et al. (2023).

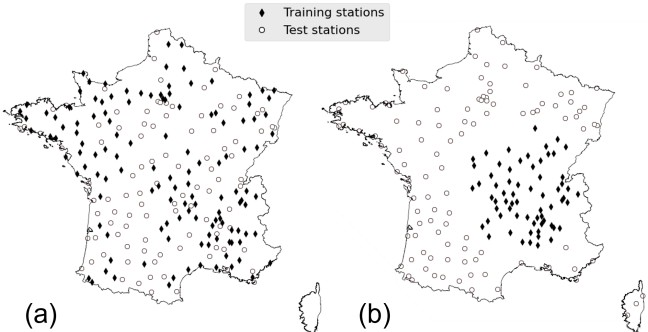

**Figure 2.** Distribution of train and test stations for training setups 1, 2, and 3 (**a**) and training setup 4 (**b**).

– Training setup 3, described in Sect. 4.2.3, allows us to understand the sensitivity of the model performance to the density of training stations when they are still intertwined with the test stations.

– Training setup 4, described in Sect. 4.2.4, enforces a geographical separation between training and test stations and thus allows us to test the ability of the model to generalize to locations geographically outside its training space.

For each training setup, a validation set is needed for early stopping (Sect. 3.3). In all four setups, 20 % of the training stations (or 10 stations if that percentage is lower) are randomly chosen in the training set to constitute the validation set.

### 4.2.1 Training setup 1

In the first setup, 100 test stations are chosen randomly from those passing QC for more than 30 % of the hours over the test period (1 July 2018 to 30 June 2019). In other words, QC must be passed for at least 8 h per day on average. As nighttime is always flagged as failing QC, this is a stringent requirement.

All the remaining stations that pass QC for more than 30 % of the hours over the training period (1 January 2010 to 30 June 2018) are used as training stations – there are 129 of them. Three techniques are further applied:

– The 100 test stations are chosen as a priority among the stations that do not pass QC for the training period. That is to maximize the number of stations used.

– The Carpentras station is manually added to the test set; that is because it is part of AERONET and BSRN (see Sect. 2.2) and can thus be used for a more thorough analysis of the models' performance.

– The three meteorological stations located in Corsica, an island 160 km from the shores of metropolitan France, are not considered in this use case.

The resulting training and test stations are shown in Fig. 2a.

### 4.2.2 Training setup 2

The second setup is identical to setup 1, except that only $Y$ years out of the 5 available are used for training, with $Y$ equal to 1, 2, 3, 4, or 5. The $Y$ years closest to the testing period are used.

The same number of training stations as in training setup 1 is used for all $Y$. However, for low values of $Y$, because some stations do not have data for the whole training period, they may only add a few points to the training set.

### 4.2.3 Training setup 3

The third setup is also very similar to setup 1. The same 100 stations make up the test set, but only $N$ stations are picked for training. $N$ varies from 20 to 100. There are $\binom{129}{N}$ ways to choose $N$ training stations among 129 candidates, and the performance of the model is likely impacted by this choice, especially with low $N$. However, training with every possible combination is not computationally tractable[5]; instead, we randomly pick 20 combinations for each $N$.

### 4.2.4 Training setup 4

In the fourth setup, we enforce geographical separation between the training and test set. All the stations within a circle centered at 46° N, 4° E, and with a radius of radius 215 km passing QC for more than 30 % of the hours over the training period (1 January 2010 to 30 June 2018) are taken as training stations, and all stations outside of a circle centered at 46° N, 4° E, and with a radius of 255 km passing QC for more than 30 % of the hours over the test period (1 July 2018 to 30 June 2019) are used as test stations. This separation is somewhat arbitrary; the objective is to keep enough stations in the training set while concentrating them in a region as

---

[5]$\max\left(\binom{129}{N}\right) \approx 4.8 \times 10^{37}$ ($N = 65$).

small as possible. This results in 66 training stations and 105 test stations, as illustrated in Fig. 2b.

### 4.3 Clear-sky detection

In Sect. 5.2, we focus our analysis on days with a majority of clear-sky instances. It is difficult to accurately detect clear-sky instances with mean hourly SSI; we, therefore, restrict the analysis to the station of Carpentras, for which we have minute data (Sect. 2.2). We first detect clear-sky minutes with a 1 min resolution, using the Reno and Hansen algorithm (with a window length of 10 min) (Reno and Hansen, 2016) implemented in the pvlib Python library (Holmgren et al., 2018). We then select days for which 75 % of the daytime is detected as clear sky.

## 5 Results

This section is divided into three parts. In Sect. 5.1, we analyze the general performance of the ML model with training setup 1 for the whole test period. In Sect. 5.2, we still work with training setup 1 but focus on the specific case of clear-sky days for the station of Carpentras. Finally, in Sect. 5.3, we discuss the impact of the number and location of training stations on the performance of the ML model (training setups 2, 3, and 4).

### 5.1 Model performance with a dense training set

In this section, we analyze the performance of the ML model with training setup 1, i.e., using the maximum number of randomly chosen training stations (129). Although training and test stations are different, they are largely interlaced (Fig. 2a).

#### 5.1.1 Overall performances

We first evaluate the ML model and CAMS performance metrics for all 100 test stations and the whole test period. The overall metrics are shown in Table 3. The ML model has a significantly lower RMSE and SDE than CAMS: 19 % and 18 %, respectively. The correlation between the ML model and the ground measurements is higher than that between CAMS and the ground measurements. In terms of bias, on the other hand, the difference between the two retrieval models is negligible: both MBEs are relatively low.

To better understand the characteristics of the ML model and CAMS estimations, we look at the joint distribution between estimations and ground measurements, shown in Fig. 3a and b. As suggested by the ML model's lower SDE, the joint distribution of this model is more tightly wrapped around the axis $x = y$ than that of CAMS. In addition, the joint distribution does not show any artifact or unphysical features – as is sometimes the case for overly smooth estimations or forecasts, for example (Verbois et al., 2020).

**Table 3.** Overall test metrics for CAMS and the ML model with training setup 1 (computed over 391 481 samples).

|  | ML model (training setup 1) | CAMS |
| --- | --- | --- |
| RMSE (W m$^{-2}$) | 52.92 | 64.99 |
| SDE (W m$^{-2}$) | 52.28 | 64.02 |
| MBE (W m$^{-2}$) | −8.21 | −11.22 |
| $\rho_{\text{pearson}}$ | 0.977 | 0.966 |

The distribution of SSI – estimated or measured – is highly dominated by the diurnal and annual pattern of the sun. To focus on the ability of the retrieval models to resolve clouds, we compare the clear-sky indices from the estimations and from the ground measurements in Fig. 3c (ML model) and Fig. 3d (CAMS) by analyzing their joint distribution. Overall, ML model estimations of the clear-sky index are more likely to be close to the ground measurements. In addition, CAMS estimations of the clear-sky index are constrained to the interval [0.1, 1] by design, whereas the ML model better matches the distribution of the ground measurements, with the clear-sky index values ranging from 0 to 1.8. Admittedly, this only concerns a small portion of all instances, and, in addition, the ML model tends to produce too many estimations with a high clear-sky index.

#### 5.1.2 Station-wise performances

Beyond the overall performance, a retrieval model needs to be consistent. We, therefore, analyze the performance of the ML model and CAMS for each test station independently. Figure 4 compares the RMSE, SDE, and MBE of the two models for each station: one point on the graph corresponds to one station, and the green band identifies stations for which the ML model is better than CAMS. We see that in terms of RMSE, the ML model outperforms CAMS for all but four stations. Furthermore, the difference between the two models for these four stations is small. In terms of SDE, the ML model does even better, as it outperforms CAMS at 98 % of the test stations. In terms of bias, interestingly, the ML model has a higher MBE than CAMS for 58 % of the stations, even though its overall MBE was lower than that of CAMS. In addition, although CAMS and the ML model have a low MBE overall, it reaches 50 W m$^{-2}$ in some locations, which is not negligible.

#### 5.1.3 Performance analysis with respect to different conditions

To complete our analysis of the two models' overall performances, we look at the metrics' dependence on the sky conditions. We use the clear-sky index as a proxy: a low clear-sky index corresponds to overcast skies, a high clear-sky index to mostly clear skies, and an intermediate clear-sky index

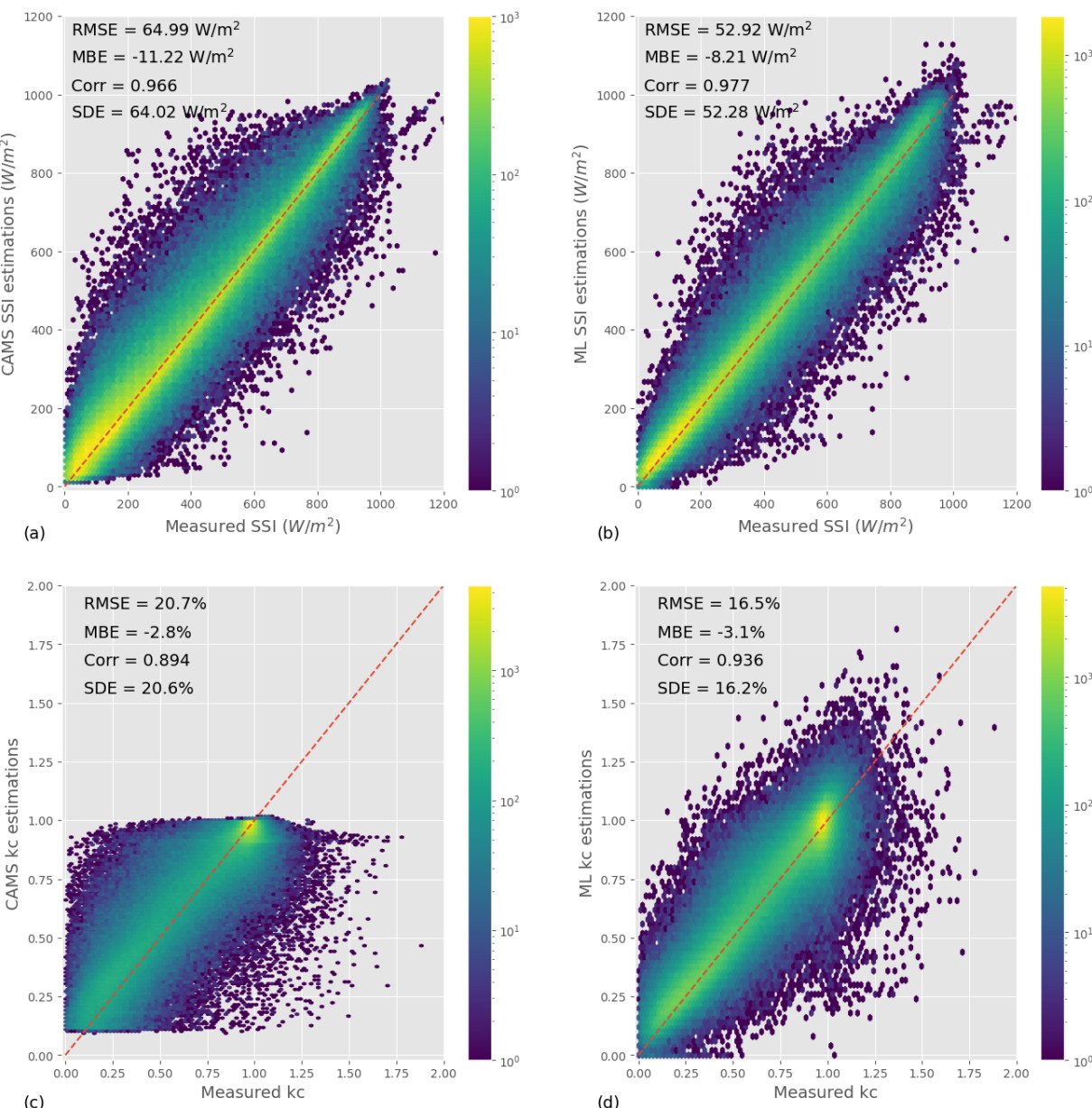

**Figure 3.** Joint distributions of satellite-derived estimations and ground measurements for the CAMS model (**a, c**) and for ML (**b, d**).

to partly cloudy skies. This is certainly an oversimplification, and a more sophisticated analysis would be required for an accurate classification of the sky conditions; having access to the hourly average of SSI, however, the value of the clear-sky index is a good first approximation. The station-wise RMSE, SDE, and MBE of the ML model and CAMS are broken down per class of clear-sky index in Fig. 5; box-plots are used to represent the metric's spread across stations (each boxplot is built with 100 points: 1 for each test station). We show in Sect. 5.1.1 that the ML model has a lower RMSE and SDE than CAMS; we see here that it is mostly for low clear-sky indices that the ML model outperforms CAMS. For clear-sky indices larger than 0.9, both retrieval models have

similar RMSE, and CAMS even has a slightly lower SDE in that clear-sky index interval. In terms of bias, although both models have similar MBE overall (Table 3), their dependence on $k_c$ is different. CAMS overestimates the SSI for low clear-sky indices and underestimates it at high clear-sky indices; the ML model, in contrast, systematically overestimates the SSI, but to a lesser extent.

## 5.2 Specific case of clear-sky days

The previous section should convince us that with training setup 1, the ML model significantly and systematically outperforms CAMS in mainland France and under all-sky con-

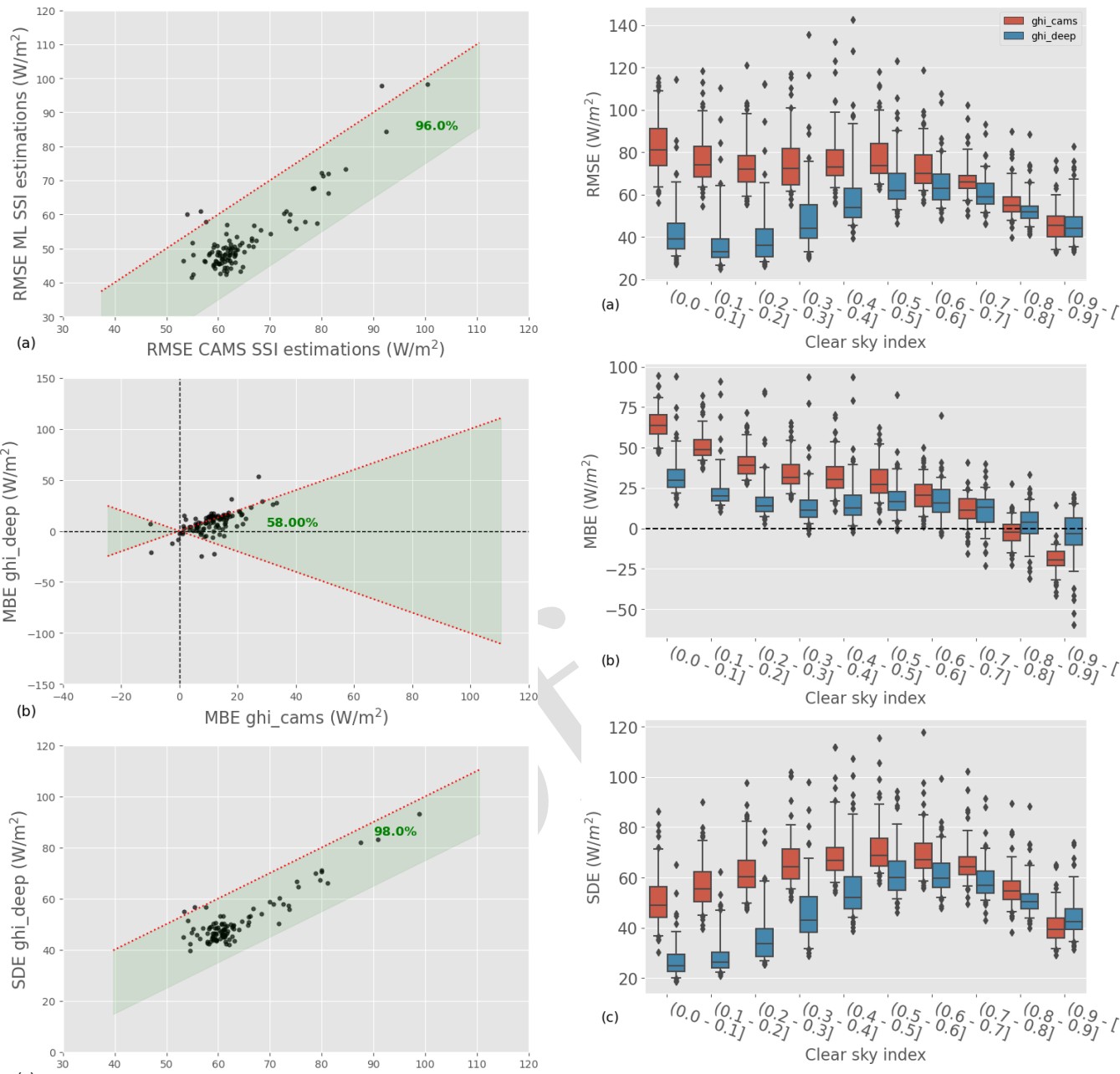

**Figure 4.** Comparison of RMSE **(a)**, MBE **(b)**, and SDE **(b)** of the ML model and CAMS for each station. The green band indicates the stations for which the ML model outperforms CAMS; the corresponding percentage is indicated in bold green.

**Figure 5.** Distribution of station RMSE **(a)**, MBE **(b)**, and SDE **(c)** of the ML model and CAMS as a function of the clear-sky index $k_c$. Each boxplot is built with 100 points: 1 for each test station.

ditions. Figures 3 and 5, however, suggest that things may be different under clear-sky conditions. Furthermore, SSI retrieval from satellite observations notably involves specific considerations when there are no clouds: aerosol concentrations and ground albedo, for example, have a stronger impact on physical estimations in cloudless skies (Scheck et al., 2016).

In this section, we focus on the performance of the two retrieval models under clear-sky conditions. To accurately identify such conditions, however, the analysis done in Fig. 5 is not sufficient: all clear-sky situations should be contained in the right-most $k_c$ bin ((0.9 − [), but other situations (typically a mix of overshooting, clear-sky conditions, and partially cloudy conditions) are likely also contained in this bin. To rigorously select clear-sky conditions, we need 1 min irradiance data (Sect. 4.3); we thus focus on the Carpentras

**Table 4.** Performance of the ML model and CAMS for Carpentras stations under all skies (4012 samples) and under clear skies only (938 samples).

|           |                          | ML model (training setup 1) | CAMS  |
|-----------|--------------------------|:---------------------------:|:-----:|
| All skies | RMSE (W m$^{-2}$)        | 42.31                       | 55.09 |
|           | MBE (W m$^{-2}$)         | −3.31                       | 7.40  |
|           | SDE (W m$^{-2}$)         | 42.18                       | 54.59 |
|           | $\rho_{\text{pearson}}$  | 0.987                       | 0.978 |
| Clear-sky days | RMSE (W m$^{-2}$)   | 21.82                       | 15.95 |
|           | MBE (W m$^{-2}$)         | −4.92                       | −2.07 |
|           | SDE (W m$^{-2}$)         | 21.27                       | 15.81 |
|           | $\rho_{\text{pearson}}$  | 0.996                       | 0.999 |

station (Sect. 2.2). Note that the ML model is the same as the one discussed in Sect. 5.1; only the analysis is restricted to one location.

### 5.2.1 Clear-sky performances

The performance metrics of the ML model and CAMS for all skies and clear-sky days are shown in Table 4. As expected, the ML model has a lower RMSE and SDE than CAM for all skies; it even has a slightly lower MBE. For clear-sky days, both models have a significantly lower RMSE and SDE. But, contrary to the general case, CAMS significantly outperforms the ML model in all metrics, with RMSE, SDE, and MBE 27 %, 26 %, and 57 % lower, respectively.

Several factors could explain the deficiency of the ML model under clear skies. In cloudless conditions, the albedo of the ground plays a more important role than under cloudy skies; since the ML model has no information about this quantity, it could be one source of uncertainty. Aerosols and in particular aerosol optical depth (AOD) are also important under clear skies; CAMS, through the clear-sky model Mc-Clear, accounts for AOD in its estimations, but the ML model has no direct access to this information.

### 5.2.2 Impact of aerosols

Albedo variations are most often more significant in space, while AOD varies in both space and time. Because we perform the clear-sky specific analysis in a single location, it is difficult to investigate the impact of albedo on the clear-sky performance. We, therefore, focus in this section on the impact of aerosol on CAMS and the ML model clear-sky estimations.

Figure 6a shows the AOD at 500 nm measured at the Carpentras station for 2 consecutive clear-sky days. We chose these dates as illustration because a significant drop in AOD can be observed from one day to the next. The corresponding measurements of hourly SSI are shown by black crosses in Fig. 6b. Even though both days have a clear-sky profile, the

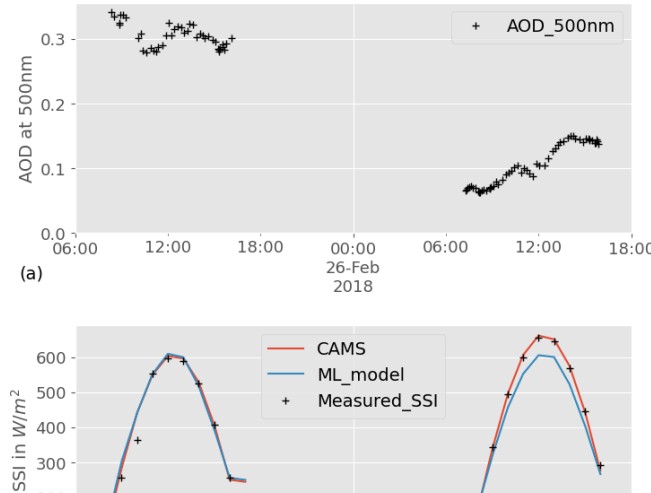

**Figure 6.** Example of 2 consecutive clear-sky days (with aerosols) Carpentras.

SSI values are significantly higher for the second day, particularly in the middle of the day. CAMS estimations of SSI for that day, shown in the same figure in red, match the observations very well: the model rightfully integrates the effect of aerosols. The ML model, in blue in the figure, correctly estimates 2 clear-sky days, but the values of SSI for the 2 d are nearly identical: as suspected, the ML model is not able to account for the effect of aerosol as well as CAMS.

To further investigate the role of information about AOD at 500 nm in the ML model underperforming for clear-sky days, we analyze the relationship between the hourly estimation error and the corresponding hourly AOD average under clear-sky conditions. This relationship is illustrated in Fig. 7, which shows the distribution of the error in each retrieval model as a function of AOD 500; the corresponding Spearman correlation is also displayed. Although there is no obvious pattern, the error in the ML model appears to have a relationship with AOD 500, as confirmed by the relatively high Spearman correlation. CAMS error, on the other hand, is weakly correlated with AOD 500. The remaining correlation may come from the fact that CAMS uses modeled AOD, which can deviate from the ground truth. Even though correlation is not causation, this result further supports the hypothesis that not accounting for AOD 500 in the ML model causes some of the estimation error under clear skies.

This result is somewhat expected, as the CAMS model integrates some information about the AOD (through Mc-Clear), whereas the ML model does not. Adding AOD-related predictors to the neural network may help decrease the performance gap between the two methods for clear skies.

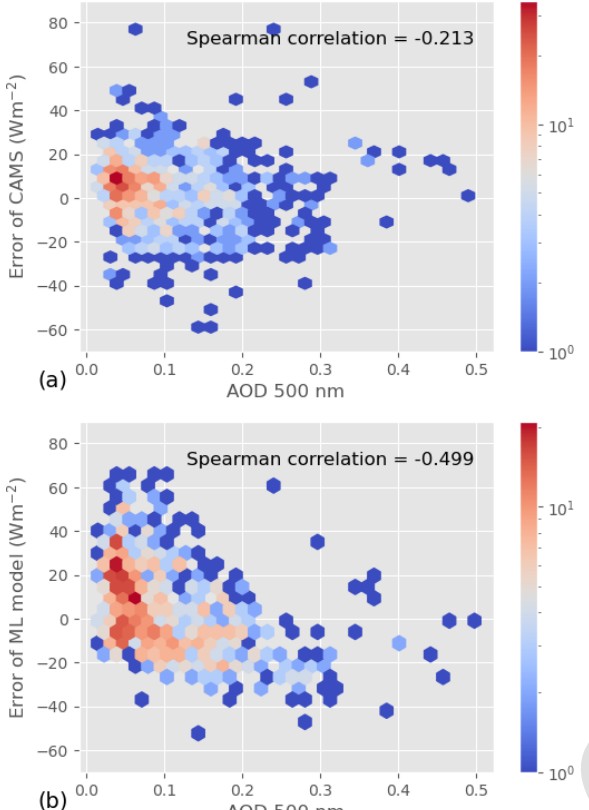

**Figure 7.** Joint distribution (2D histogram) of hourly average AOD and hourly estimation error for CAMS **(a)** and the ML model **(b)**. Spearman's rank-order correlation between AOD and error is also given.

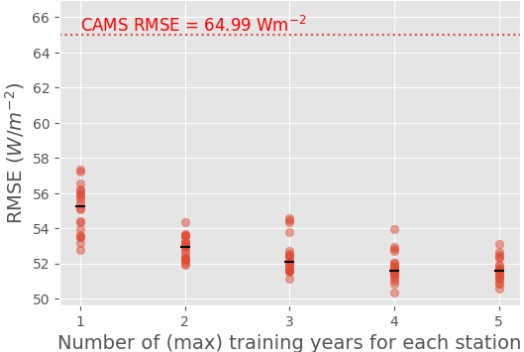

**Figure 8.** Test RMSE as a function of the number $Y$ of years used for the training (training setup 2); 20 models were trained for each $Y$ to account for the variations due to random initialization. Each red point represents the RMSE for 1 of the 20 models; the median performance for each $Y$ is shown by a black line.

## 5.3 Sensitivity to the training set

To this point, we have analyzed the performance of the ML model with training setup 1, i.e., when the neural network is trained with 129 stations, interlaced with the test stations (Fig. 2a). Such a density of measurement stations is rare, and many of the regions covered by MSG – and thus by CAMS – are not as well equipped. In this section, we therefore evaluate the impact of the size and location of the training set on the performance of the ML model. We first reduce the number of training years (training setup 2; Sect. 4.2.2) and training stations while keeping the random split (training setup 3; Sect. 4.2.3) and then enforce geographical separation between training and test stations (training setup 4; Sect. 4.2.4). In this section, we focus on RMSE for conciseness.

### 5.3.1 Impact of the number of training years

We first evaluate the impact of the number of training years on the ML model's performance using training setup 2 (Sect. 4.2.2). Figure 8 shows the RMSE of the ML model for the 100 test stations when the model is trained with all 129 training stations but with a different number $Y$ of train-

ing years. To account for the variations due to the model's random initialization (further discussed in Appendix B), 20 models were trained for each $Y$. The median RMSE is shown by a black line for each $Y$. Interestingly, the variations due to random initialization of the network are more important than the variations due to the number of training years, making the interpretation a bit uncertain. The performance of the ML model nonetheless appears slightly impacted by the number of training years: the median RMSE of the ML model decreases monotonously with the increasing number of training years, with a maximum of $55\,\mathrm{W\,m^{-2}}$ for $Y = 1$ and a minimum of $52\,\mathrm{W\,m^{-2}}$ for $Y = 5$. For $Y \geq 3$, however, the improvement is negligible.

Importantly, the ML model performs significantly better than CAMS, even with a single training year. One year of data for 129 stations is a relatively large dataset; it is, therefore, not surprising that it suffices for the small neural network used here (a MLP) to converge. However, it is noteworthy that the diversity of situations encountered with 1 year and 129 stations is sufficient for the ML model to largely outperform CAMS.

### 5.3.2 Impact of the number of training stations

We then consider the influence of the number of training stations on the ML model performance using training setup 3 (Sect. 4.2.3). Figure 9 shows the RMSE of the ML model for the 100 test stations as a function of the number of training stations $N$. As discussed in Sect. 4.2.3, we repeat the experiment 20 times for each choice of $N$, with different randomly chosen training stations at each iteration. For $N = 129$, as there are only 129 candidates, the 20 iterations are done with the same training stations. The variations in the RMSE for $N = 129$ are thus only caused by the variations in the random initialization of the weights between runs, discussed in Appendix B.

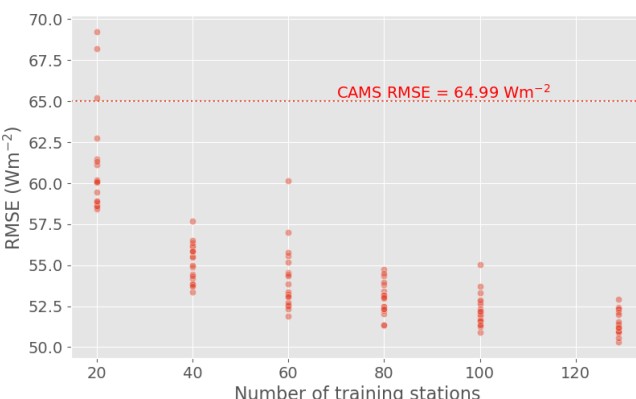

**Figure 9.** Test RMSE as a function of the number $N$ of stations used for the training with 20 random picks of $N$ among 129 (training setup 3).

For $N \geq 40$, the RMSE of the ML model remains significantly lower than that of CAMS, even though it increases a bit on average for $N \leq 100$. For $N = 20$, however, the ML model performances deteriorate markedly: the RMSE of the best-performing run is higher than for $N \geq 40$, and the RMSE of the worst-performing run largely exceeds that of CAMS. We also notice that the RMSE variations between runs are more important for $N = 20$. Put in perspective with the results of Sect. 5.3.1, this suggests that the issue is not the size of the training set, but the location of the training stations.

### 5.3.3 Impact of the location of the training stations

To further investigate the impact of the relative location of the training and test stations on the ML model's performance, we enforce a geographical separation between them (training setup 4; Sect. 4.2.4)[6]. Table 5 shows the overall metrics for the ML model and CAMS. The performance of the latter is similar to the one described in Sect. 5.1, even though we use different test stations. Contrastingly, the RMSE and SDE of the ML model are much higher with this training setup than they were with training setup 1 or 2 (with $N \geq 40$). Whereas the ML model outperformed CAMS with training setup 1, the average performances of the two retrieval models are almost equivalent here.

As in Sect. 5.1.3, it is interesting to analyze the performances per station. Figure 10 compares the RMSE of the ML model and CAMS for each station. We see that while the two retrieval models have similar RMSE on average, the distributions of the station-wise RMSE are very different. The ML model's RMSE is slightly lower for 82 of the 105 test stations, while CAMS performs somewhat better for 18 other stations. For three to five locations, however, the ML

---

[6]It should be noted that the test stations are not the same as in training setup 1 and 2; the values of RMSE, MBE, SDE, and $\rho_{\text{pearson}}$ should thus not be compared with previous sections.

**Table 5.** Overall test metrics for CAMS and the ML model with training setup 4 (computed over 411 733 samples).

|  | ML model (training setup 4) | CAMS |
| --- | --- | --- |
| RMSE (W m$^{-2}$) | 61.04 | 63.49 |
| MBE (W m$^{-2}$) | −6.69 | 10.25 |
| SDE (W m$^{-2}$) | 60.68 | 62.66 |
| $\rho_{\text{pearson}}$ | 0.969 | 0.967 |

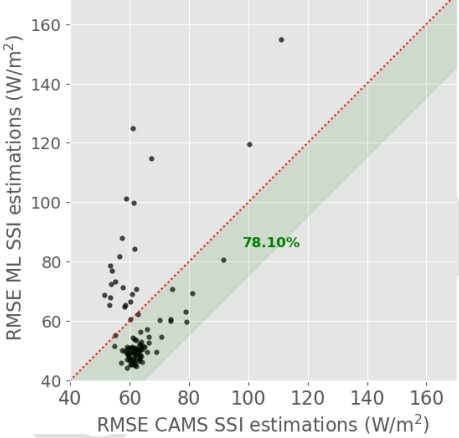

**Figure 10.** Comparison of RMSE of the ML model (training setup 4) and CAMS for each station. The green band indicates the stations for which the ML model outperforms CAMS; the percentage of such stations is indicated in bold green.

model's RMSE is dramatically higher than that of CAMS: in the worst case, $\text{RMSE}_{\text{ML model}}$ is more than 2 times higher than $\text{RMSE}_{\text{CAMS}}$.

### 5.3.4 Impact of albedo

We have shown that, with geographical separation between training and testing sets (training setup 4), the ML model performs reasonably well on average but is susceptible to providing highly inaccurate estimations in some locations. To try and understand what causes highly inaccurate estimations, the geographical distribution of test RMSE skill is represented in Fig. 11. Interestingly, the distance to the training set does not have a clear impact on the performance of the ML model. Rather, most of the stations for which the ML model is largely outperformed by CAMS (i.e., with high negative skill scores) are located on the Mediterranean or Atlantic coasts. Ocean and continental tiles have different albedos, which significantly impacts the radiance observed by the satellite (Blanc et al., 2014). Physical retrieval models account for that difference, but the ML model does not have direct access to that information; this could explain its poor performance in seaside stations.

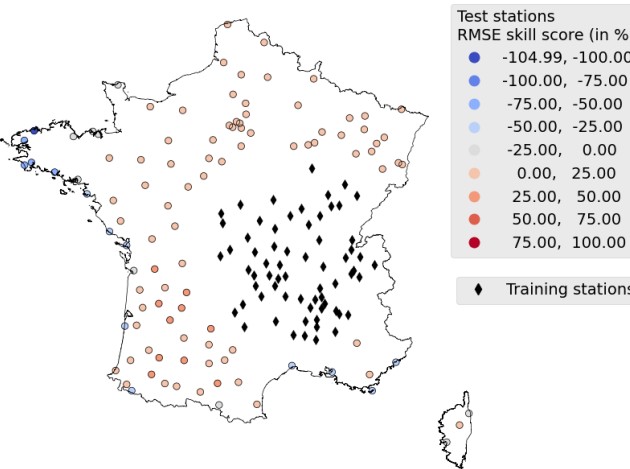

**Figure 11.** Geographical distribution of the RMSE skill score of the ML model in training setup 4. Positive values show the stations where the ML model outperforms CAMS.

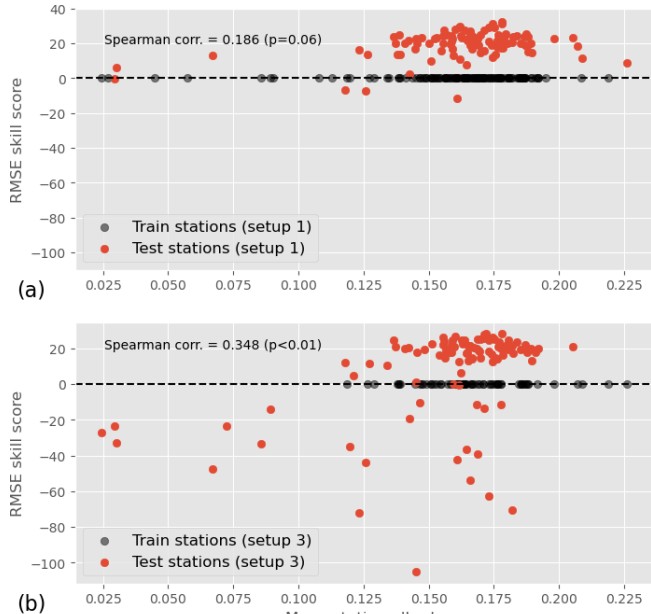

**Figure 12.** RMSE skill score as a function of station mean albedo for training setup 1 (**a**) and training setup 4 (**b**). The distribution of the training stations' albedo is also shown on the $y = 0$ axis.

To test that hypothesis, Fig. 12b shows the RMSE skill score of each test station as a function of its mean albedo. We observe that while a negative skill score does not necessarily imply a low albedo, a low albedo (lower than 0.1) systematically comes with a negative skill score. The existence of a positive relationship between albedo and RMSE skill score is further supported by a statistically significant Spearman correlation coefficient of 0.348 between the two values. Figure 12a shows the same plot but for training setup 1. We see that in this case, low albedo does not come with negative RMSE skill scores. This absence of a relationship – or at least its lower strength compared to training setup 4 – is confirmed by a statistically non-significant ($p = 0.07$) Spearman correlation coefficient of 0.184 between the RMSE skill score and the albedo.

To understand the difference in behavior between the two training setups, it is useful to look at the distribution of the albedo for training stations in either case; the mean albedos of training stations are hence shown on the $y = 0$ axis in Fig. 12. In training setup 1, several training stations have a mean albedo between 0.025 and 0.1, while in training setup 4, all training stations' albedos are greater than 0.1. This suggests that the RMSE skill score is not directly influenced by the test station albedo, but rather by the distance between the test station albedo and the training station albedo. In other words, the ML model is not able to generalize to stations with an albedo it has not seen during training.

## 6 Discussions and conclusions

### 6.1 Great potential with a dense training set despite some caveats

Machine learning for satellite retrieval has great potential. Provided we have the right data, performance improvement over traditional approaches can be important. We indeed showed that when trained with a network of measurement stations spread evenly across France, a simple neural network has significantly lower error metrics and better overall representativity than CAMS, a state-of-the-art physical retrieval model. Because we ensured that we tested the ability of the model to extrapolate in space and time, that means that such a model could be used operationally and, on average, provide better estimations than CAMS.

We found, however, that the neural network is not able to properly account for the role of aerosols in clear-sky estimations, whereas the underlying CAMS model – as well as other physical models – can. This only slightly impacts the performance of the ML model in France, where the effect of AOD on SSI is relatively small, but in other regions – for example desertic zones (Eissa et al., 2015) – the ML model may underperform. Perhaps more critically, this lack of representativity of physical phenomena undermines the confidence in the model.

## 6.2 Strong dependence on the training set

Our results show that the model's performance is very dependent on the training set. First, we found that even a simple network – with only one hidden layer – requires a relatively large number of training stations to outperform CAMS. In many regions, good-quality ground measurements are too scarce for this model to be useful. Therefore, while the ML model tested in this work could easily be adapted to be used operationally in France, it is unlikely that it can be extended to most other regions of the globe.

We further demonstrated that rather than the number of training stations, their location relative to the test sites is crucial. Our analysis showed that, in certain configurations, the neural network can underperform even at stations located close to the training set. We know that neural networks often have difficulty making predictions out of the training domain; the challenge here is that determining which location is out of the training domain is not straightforward. Whether two locations are similar in the eye of the network does not depend directly on the geographical distance between these locations. Our analysis suggests that its albedo may play a role in the ability of the neural network to generalize to a location, but it is likely not the only cause. Understanding the factors that describe the similarity between two locations should be an important aspect of future research.

## 6.3 Perspectives

Third-generation geostationary satellites are already operational above the United States (GOES-R) and Japan (Himawari-8), while Meteosat Third Generation will soon cover Europe and Africa. These new meteorological satellites have better temporal, spatial, and spectral resolutions than their second-generation counterparts. They thus produce a significantly larger amount of data. To treat these data operationally and fully benefit from the additional information, deep learning certainly has a critical role to play.

However, the solar research community needs to address the limitations of purely statistical models, as revealed in this paper. We believe that the answer resides at least partly in hybrid models, mixing physical modeling and statistical learning. Variations in hybrid models include the use of machine learning models trained on datasets derived from physical simulations. These models can serve as proxies for parts of existing physical models and can be further fine-tuned on real datasets via transfer learning. This approach balances the incorporation of underlying physical principles with considerations of real-world complexities and uncertainties. Another approach is to design machine learning models with physical constraints incorporated as regularization, such as conservation laws and material properties. This can ensure that the model stays within the realm of physical possibility while also incorporating data-driven components. A third option could be the direct incorporation of physical equations into the loss function of the machine learning model. This approach optimizes the model's predictions to be both data-driven and physically consistent. During the training process, the model is guided by both observed data and underlying physical laws.

A better understanding of the generalization capabilities of the models is also critical. We see in this paper that the albedo may play a role, but more research is needed to understand the extent to which and the conditions under which we can expect the model to generalize well. Data segmentation algorithms could be useful to optimize the construction of training datasets and to identify locations where the retrieval model may not be trusted.

The investigation of more sophisticated neural network architectures is also of interest and would become particularly relevant when dealing with larger input datasets. Architectures such as convolutional neural networks (CNNs), recurrent neural networks (RNNs), or spatio-temporal transformers hold promise, especially when a broader context in both time and space is required. However, it is important to recognize that such complexity may raise the risk of generalization issues, as more complex models are generally more likely to overfit.

Finally, we must remember that machine learning models are often opaque, making it difficult to understand how they make their predictions. This means that it is unlikely, at least in the short term, that we will be able to derive new physics from these models. If we focus only on machine learning, we may limit our understanding of the world around us. We, therefore, believe that the research community should continue to invest in the development and improvement of physical retrieval models.

## Appendix A: Quality check

The quality check procedure applied to Météo-France ground measurements is described in detail in Verbois et al. (2023). In summary, it consists of the following checks.

1. Each value is tested for *extremely rare limits* (ERLs) as recommended by Long and Dutton (2010): $-2 < \text{GHI} < 1.2 I_{\text{sc}} \cos^{1.2}(\theta_{\text{z}}) + 50 \,\text{W m}^{-2}$, where $I_{\text{sc}}$ is the solar constant adjusted for Earth–sun distance, and $\theta_{\text{z}}$ is the solar zenith angle.

2. A digital model of the horizon (Blanc et al., 2011b) is used to exclude every instance for which the sun is under the horizon.

3. A visual check for the spatial coherence of $k_{\text{c}}$ is performed.

4. A visual check for shadows is performed in the solar azimuth–solar elevation plane.

Out of the original 286 stations available, 46 are fully excluded by QC.

## Appendix B: Performance variations due to random initialization

Figure B1 shows the RMSE, MBE, and SDE of 20 models run with exactly the same setup, described in Sect. 3.3, but a different randomly chosen weight initialization. SDE and RMSE vary between 50–52 and 50–53 W m$^{-2}$, respectively; this is relatively small compared to CAMS, which has an RMSE of 64.9 W m$^{-2}$ and an SDE of 64 W m$^{-2}$. The MBE, on the other hand, varies between 0 and 9 W m$^{-2}$. That is more important compared to CAMS MBE (ca. 12 W m$^{-2}$), but still relatively small compared to the average SSI in France.

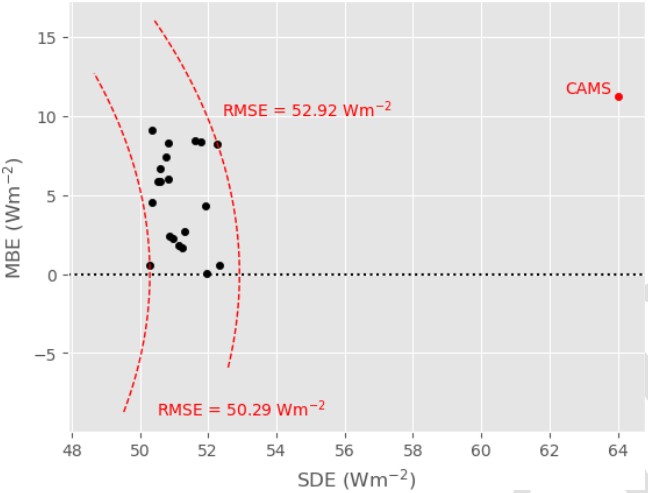

**Figure B1.** Target diagram showing the RMSE, MBE, and SDE of 20 models run with exactly the same setup, but a different weight initialization.

*Data availability.* The following data sources are accessible online for free.

- CAMS estimates of solar surface irradiances and clear-sky irradiances can be downloaded from the SoDa website (https://www.soda-pro.com/help/cams-services/cams-radiation-service/download-europe-volume TS3) or with the following pvlib function: https://pvlib-python.readthedocs.io/en/stable/reference/generated/pvlib.iotools.get_cams.html TS4.
- MSG data are available on the EUMETSAT website: https://navigator.eumetsat.int/product/EO:EUM:DAT:MSG:HRSEVIRI TS5.
- Ground irradiance data for the Carpentras station can be downloaded from the BSRN website: https://bsrn.awi.de (last access: 5 September 2023; https://doi.org/10.1594/PANGAEA.896713, Brunier, 2018).

Météo-France data were generously provided by Météo-France for research purposes. More information can be found on the Météo-France public data website: https://donneespubliques.meteofrance.fr (last access: 5 September 2023).

*Supplement.* The supplement related to this article is available online at: https://doi.org/10.5194/amt-16-1-2023-supplement.

*Author contributions.* HV, YMSD, and PB were responsible for conceptualization and methodology. YMSD and BG handled data curation. HV and BG were responsible for software. HV and VB performed formal analysis and visualization. HV prepared the original draft. YMSD, VB, BG, and PB reviewed and edited the paper.

*Competing interests.* The contact author has declared that none of the authors has any competing interests.

*Acknowledgements.* The work of Hadrien Verbois, Yves-Marie Saint-Drenan, Vadim Becquet, and Philippe Blanc was supported by the SciDoSol chair.

*Financial support.* This research has been supported by the SciDoSol chair. TS6

*Review statement.* This paper was edited by Sandip Dhomse and reviewed by four anonymous referees.

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

**Remarks from the language copy-editor**

CE1     Please note that "generalize" is a transitive verb. As such, the current phrasing implies that the models cannot generalize something else to locations they have not seen during training. If you mean the models cannot be generalized to these locations, this should be changed. Please confirm for all instances and forms of the verb "generalize" throughout the text.

CE2     You requested Meteosat Second Generation, but it appears you meant Meteosat Third Generation. Please confirm.

**Remarks from the typesetter**

TS1     Due to the requested changes, we have to forward your requests to the handling editor for approval. To explain the corrections needed to the editor, please send me the reason why these corrections are necessary. Please note that the status of your paper will be changed to "Post-review adjustments" until the editor has made their decision. We will keep you informed via email.

TS2     Please check.

TS3     If the website is maintained by a company, the company serves as the creator, so e.g. "Solar radiation data: Europe Volume "Agate", MINES ParisTech/Transvalor [data set], https://www.soda-pro.com/help/cams-services/cams-radiation-service/download-europe-volume, last access: 5 September". Please check if this reference can be used and provide similar references for the other links.

TS4     Please provide a reference list entry including creators, title, repository/publisher, and date of last access.

TS5     Please provide a reference list entry including creators, title, repository/publisher, and date of last access.

TS6     Thank you for providing this information. Please note that we allow the funding information to be included in both the acknowledgements and the financial support section if you would like to leave the acknowledgements section as it is, or this information can now be removed from the acknowledgements. Please let us know how you would like to proceed. Thank you.

TS7     This reference is not cited in the text anymore (due to correction on page 4). Please check.

TS8     Please confirm initials.

TS9     Please provide DOI.

TS10     Please provide date of last access.

TS11     Please provide a persistent identifier (ISBN or DOI preferred).

TS12     Please provide initials.

TS13     Please provide a persistent identifier (ISBN or DOI preferred).

TS14     Please provide exact date.

TS15     Please provide date of last access.

TS16     Please provide publisher.

TS17     Please provide all author names and make sure that all authors are listed in the correct order: last name, initial(s).

TS18     Please note that the year has been adjusted here and in the text.

TS19     Please provide date and location of the workshop.