# Peer review of "Retrieval of surface solar irradiance from satellite using machine learning: pitfalls and perspectives"

_EGUsphere, 2023_

## Author Comment (AC1)

**GENERAL COMMETS:**

1. I think this manuscript addresses a very important point, the pitfalls and drawbacks of AI for the retrieval of SSI. For example, the analysis of the results at the Mediterranean stations presented in this manuscript illustrates the challenges associated with machine learning. The net is only able to learn from local relations between reflection, surface albedo, atmosphere and SSI. It knows nothing about physics. Hence, it has to be expected that the performance decreases significantly if it is applied in regions with quite different conditions concerning aerosol load, cloud types, H20 and surface albedo. Within this scope it has to be taken into account that in many regions almost no in-situ data are available for the training or retraining. Hence, no learning of regional relations is possible then, but physical retrieval methods do not have these problems and it would be interesting to see the results between the current network with CAMS in Africa. Hence, the question why AI is needed for the retrieval of SSI should be addressed in more detail in the manuscript..

   The results presented in this work strongly suggest that such a model is NOT applicable to regions with less dense measurement networks. We had already suggested this in the conclusion ("In many regions, good quality ground measurements are too scarce for this model to be useful."). To make it clearer, we added the following:

   > "Therefore, while the ML model tested in this work could easily be adapted to be used operationally in France, it is unlikely that it can be extended to most other regions of the globe."

   Further, it is difficult to know what the net has really learned (black box approach),  If we do not know what the net learns, we can't learn either  (and our intelligence might expire on the long run). This should be discussed in more detail as well, based on the results presented in the manuscript. These points are partly addressed in the conclusion (e.g. L370ff) but should be discussed in more detail.

   We fully agree with the remark about the "black box" approach. We have added a new paragraph to the conclusion:

   > "Finally, we must remember that machine learning models are often opaque, making it difficult to understand how they make their predictions. This means that it is unlikely, at least in the short term, that we will be able to derive new physics from these models. If we focus only on machine learning, we may limit our understanding of the world around us. We therefore firmly believe that the research community should continue to invest in the development and improvement of physical retrieval models."

2. Poor performance of AI could also result from wrong training or training architecture. However, the comparison with the established CAMS shows that the training has been done well (5.1.1). This is very good, because it shows that the discussed pitfalls are not due to failures in the training or the chosen training method.

3. Figure 5: It seems to me that the main benefit of the machine learning is that it corrects differences in SSI induced by the different viewing geometries of ground based and satellite observations. We are aware of this effect, as significant differences are apparent when SSI retrieved from Meteosat East is compared with those from Meteosat prime for the same regions. So far these effects are not considered in many physical methods e.g. in CAMS, but it might be possible to implement appropriate "slant column" geometry corrections, which would increase the comparability of ground-based and satellite-based SSI. Please discuss this issue.

   It is possible that the better performance of the ML model (in training setup 1) stems from its ability to handle different viewing geometry. However, our results cannot confirm or infirm this hypothesis. Because we use a simple neural network, however, it is in our opinion unlikely that the model is able to correct for e.g. the parallax effect.

   A more thorough investigation would be necessary to determine how sun and satellite geometry is handled by the network. We believe that this would be a very interesting topic for future work. Such a study would also need to look into recent improvements in physical models that also account for viewing and solar angle.

4. Please consider that other physical retrieval methods might perform better or worse than CAMS, hence that the network might have a lower/higher performance when compared with other models. Please mention this briefly.

   We mentioned this in section 2.3:

   "It should be noted that other physical retrieval methods might outperform CAMS (Forstinger et al., 2023). It remains, nonetheless, a state-of-the-art retrieval model."

5. 5.2.2 Impact of aerosols: This is not a really a fair analysis, AOD (and H20) have not been given at predictors for the learning, hence the network could not learn anything about the relation of AOD variations and SSI, SAT reflection. It can just learn locally some kind of mean clear sky state. Contrarily AOD is used in CAMS as "predictor". Please mention that the performance of ML might be better if AOD data were used as predictor in addition. Of course, it is not easy to find an accurate AOD raster data set, but this problem concerns AI as well as phyical

methods. Further, here, AOD from Aeronet is used, which is not available for CAMS elsewhere. Hence the capability of CAMS (or any other sat retrieval) concerning AOD variations is probably much lower as in the example. This should be also mentioned.

We added a discussion to the section 5.2.2 impact of aerosols:

"This result is somewhat expected, as CAMS model integrates some information about the AOD (through McClear), whereas ML model does not. Adding AOD-related predictors to the neural network may help decrease the performance gap between the two methods for clear-skies."

Regarding the fact that CAMS uses AOD raster data and not actual measurements, we added a footnote: "The remaining correlation may come from the fact that CAMS uses modeled AOD, that sometimes deviate from the truth."

6. Please add more information about the in-situ data. Do they all have the same maintenance, calibrations cycles and so on. Hence can the same accuracy be expected for all pyranometers ?

Unfortunately, we only have access to limited information about this. The measurement stations used in this work are all operated by Météo-France which performs regular checks and calibration. However, to the best of our knowledge, there is no synthetic information describing the schedule of these procedures.

This was a motivation for the thorough and conservative QC applied to these stations, summarized in Appendix and thoroughly described in https://doi.org/10.1016/j.solener.2023.04.037

7. Throughout the manuscript. Please avoid the separation between physical methods and clear sky index methods. They are physical methods as well !

Done!

**DETAILED COMMENTS:**

1. Abstract: "the first of which is likely solar energy". This depends on the viewpoint. Please delete "the first of" and rephrase accordingly, it is also quite important for climate, tourism,...

This was removed.

2. Abstract: "For long, the emphasis has been on empirical models (simple parameterization linking the reflectance to the clear-sky index) and on physical models" The use of the clear sky index follows also physical laws, hence please rephrase. Please see also the general comments.

We changed to:

> "For long, the emphasis has been on models grounded in physical laws with, in some cases, simple statistical parametrizations."

3. L25 "…index methods without explicit physical cloud models", L29 "empirical" please see 2.) and general comments . The use of the clear sky index follows also physical laws and the cloud index is a measure for the cloud transmission, thus, not without physical cloud model, please rephrase

   We changed to:

   > "from the earlier cloud index methods (Cano et al., 1986; Rigollier and Wald, 1998) to more recent approaches relying on advanced radiative transfer models (Xie et al., 2016; Qu et al., 2017). »

4. Line 55 "z. 4 by 5 km". it might be closer to 3.2x5.5 please check.

   There was a mistake. According to "MSG Level 1.5 Image Data Format Description (figures 10 and 11)", it is actually ca N-S 6 km and E-W 4 km. We updated the text:

   > "MSG channels have a temporal resolution of 15 minutes and a spatial resolution of 3 km at Nadir (0,0)[1], which above France corresponds to pixels of ca. 4 by 6 km (in the E-W and N-S directions, respectively) (EUMETSAT, 2017)"

5. L104: "ML model must be fully online" Please explain why ?

   This is for the comparison with CAMS to be fair – although it is currently only available after a certain delay, CAMS only uses past and present data to deliver estimations. We change 'online' to 'real-time', to make it clearer clearer.

6. L 195 "Three tricks are applied:" Please use a more appropriate term instead of tricks.

   We changed to "Three techniques are further applied:"

7. L 370 please consider to add surface albedo here

   We already mention albedo at the end of 6.2 – we intentionally left it out of the first sentence (formerly L370), because as far as we know, it is only one of the factors that impact generalization. We, however, added a sentence to insist on the need to understand which factors impact generalization:

   > "Understanding the factors that describe the similarity between two locations should be an important aspect of future research."

8.  L 385: Another option is to improve the physical methods, without AI e.g. as demonstrated by HelioMon. The accuracy of HelioMont is already close to that of BSRN stations, why fuss with AI ? Please consider to add this option to the manuscript. In the Alps it is questionable if any network would be able to learn the complex relations for all regions, because taking the spatial heterogeneity into account there are not enough ground stations.

    We agree that the community should keep increasing physical models. In response to your first comment, we added a paragraph in that sense in section 6.3:

    "Finally, we must remember that machine learning models are often opaque, making it difficult to understand how they make their predictions. This means that it is unlikely, at least in the short term, that we will be able to derive new physics from these models. If we focus only on machine learning, we may limit our understanding of the world around us. We, therefore, believe that the research community should continue to invest in the development and improvement of physical retrieval models."

---

## Author Comment (AC2)

This is a very interesting paper that deals with limitations and perspectives for the calculation of surface solar irradiance (SSI) using machine learning techniques.

The paper deals with an aspect including a number of more or less .. easy to explain, sources of errors and uncertainties. The work is high level and ends up in a publication with unique in my opinion results worth being published in AMT.

Some comments towards manuscript improvement

Abstract

At the moment the abstract is a bit like a general discussion and some metrics there, especially summarized comparisons of ML and CAMS, could be useful for a reader that will be intrigued to read more about it.

We added several metrics from the result section in the abstract:

> "We found that the data-driven model's performance is very dependent on the training set.
>
> Provided the training set is sufficiently large and similar enough to the test set, even a simple MLP has a root mean square error (RMSE) that is 19% lower than CAMS and outperforms the physical retrieval model in 96% of the test stations.
>
> On the other hand, in certain configurations, the data-driven model can dramatically underperform even in stations located close to the training set: when geographical separation was enforced between the training and test set, the MLP-based model exhibited an RMSE that was 50% to 100% higher than that of CAMS in several locations."

Introduction

What I am missing is some basic state of the art of current datasets (including CAMS) and their performance evaluation.

We updated the first paragraph of the introduction to discuss this aspect:

> "Some of these retrieval algorithms are operational and provide SSI estimations worldwide. For example, HelioClim3 (Blanc et al., 2011a) offers real-time estimations of the Global Horizontal Irradiance (GHI) over Africa and Europe. CAMS, the Copernicus Atmosphere Monitoring Service, is another near real-time service that derives SSI estimations from data collected by both Meteosat and Himawari satellites; it covers areas including Africa, Europe, and

a significant portion of Asia (Schroedter-Homscheidt et al., 2016). In the United States, the National Solar Radiation Database (NSRDB, Sengupta et al. (2018)) serves as a valuable resource, providing SSI estimates primarily from the GOES satellites. The performances of these solar radiation databases vary with the location and sky conditions; they are discussed in detail in Forstinger et al. (2023)."

Data

AERONET does not provide AOD every minute and also in cloudy days, so some clarification could be included as a short paragraph in 2.2 e.g. how AERONET data used , which wavelength for aerosol optical depth used etc.

We have updated the paragraph describing AERONET measurements:

"As an AERONET station, it provides measurements of spectral aerosol optical depth (AOD). The AOD at different wavelengths are measured with a Sun photometer but are only valid under clear-sky conditions. Cloud screening is thus applied to the raw data and measurements are therefore only available intermittently (Giles et al., 2019). In this work, we use the AOD at 500 nm."

Section 3

It is impressive the choice of using 3 hours (12 instants) as a basic hierarchy of the method. Could you explain how this choice has been decided? isn't it 3 hours relatively .. long ?

3 hours was chosen intentionally a bit long because the MLP algorithm should be able to handle redundant or irrelevant inputs. It is indeed questionable whether 12 past instants are really needed for this algorithm; similarly, it is possible that the algorithm could benefit from a larger spatial neighborhood. The two aspects are actually very likely linked.

A rigorous ablation study could help clarify this point and, as a matter of fact, several other aspects of the algorithm could be further optimized. But we think that this would be the focus of a different paper.

The 3 set ups could be of course more complex but I personally find the choice really appropriate here.

I am a bit puzzled by the  fact  that the kc=1 limitation of CAMS does not have a more visual impact on the statistics. Or is it a major factor of the ML better performance ?

It is very likely not a major factor of the ML better performance as, on the contrary, we see in Figure 5 that the MLs model does not perform that well for kc>0.9.

The reason why it does not have a more visual impact on statistics (and may seem somewhat contradictory with the clear-sky results discussed later) is that there are only a few instants for which the measured clear-sky index is above 1.

We added a sentence in section 5.1.1 to remind the reader of this:

> "Admittedly, this only concerns a small portion of all instants, and, in addition, ML model tends to produce too many estimations with high clear-sky index."

Figure 5: based on the definition given in lines 80 – 85 and the aerosol issues discussed after fig. 5 there should be clear sky index higher than 1 not visible in the figure.

This was indeed a mistake in the labeling of the bins. The last kc bin is 'open'. We have updated the plot to reflect that.

Aerosols: It is clear that the ML inputs does not include any aerosol information so figure 7 is more or less expected. A very rough predictor including an aerosol climatology (more in summer less in winter) would for sure improve this negative correlation shown in fig. 7 . Especially because this has an impact on "high solar irradiance" cases.

Indeed this result could be expected, and it is likely that adding aerosol-related data to the ML model predictor would improve its performance. We added a discussion to Section 5.2.2:

> "This result is somewhat expected, as CAMS model integrates some information about the AOD (through McClear), whereas ML model does not. Adding AOD-related predictors to the neural network may help decrease the performance gap between the two methods for clear skies."

Fig. 7 needs a bit more explanation as it is not clear if the points are based on instants, hourly or daily values.

We updated the text and the Figure caption:

> "To further investigate the role of information about AOD at 500 nm in ML model under-performing for clear-sky days, we analyze the relationship between the hourly estimation error and the corresponding hourly AOD average under clear-sky conditions. This relationship is illustrated in Figure 7, which shows the distribution of the error of each retrieval model as a function of AOD 500;"

> "Figure 7. Joint distribution (2D histogram) of hourly average AOD and hourly estimation error for CAMS (a) and ML model (b). Spearman's Rank-Order Correlation between AOD and error is also given."

I find difficult to understand how the ML can outperform CAMS for clear skies in the related bins of fig. 5 and still have these aerosol related aspects shown in fig. 7.

That is because the right-most bin of kc in figure 5 does not contain only clear-sky situation. These are hourly values of kc, so many points likely see a mix of partially cloudy and clear sky. This shortcoming of the kc binning was our main motivation to select 'true clear sky instants' in Carpentras.

We have updated the text at the beginning of section 5.2 to insist on this point:

> "In this section, we focus on the performance of the two retrieval models under clear sky conditions. To clearly identify such conditions, however, the analysis done in Figure 5 is not sufficient: all clear sky situations should be contained in the right-most kc bin ([0.9 − [), but other situations (typically a mix of overshooting, clear sky and partially cloudy) are likely also contained in this bin. To rigorously select clear-sky conditions, we need 1-minute irradiance data (Section 4.3); we thus focus on the Carpentras station (Section 2.2)."

Maybe the authors could discuss:

In general it is understandable that the paper does not introduce a method to be used in different areas but it is a kind of sensitivity study on the ML performance. For this case a really unique dataset is used with a huge number of stations. However, it would be nice to comment on perspectives of an actual application of such system. Indirectly this study can assess some kind of realistic cases of limited or not, ground-based data available that can be used for applying such methods in different areas.

This was already a bit discussed in section 6.2:

> "In many regions, good quality ground measurements are too scarce for this model to be useful."

We agree that it is an important point and to make it clearer we added the following:

> "Therefore, while the ML model tested in this work could easily be adapted to be used operationally in France, it is unlikely that it can be extended to most other regions of the globe."

The whole France and so many stations is a huge area, but still could be very different than another area with different cloud/aerosol conditions which the same results with the same number of stations and analysis can vary. E.g. aerosol (not captured) effects in N. Africa will have a crucial effect on the statistics as well as areas with different and more clouds.

We have updated section 6.1 to mention the fact that AOD are more impactful in other regions:

> "This only slightly impacts the performance of the ML model in France, where the effect of AOD on SSI is relatively small, but in other regions – for example deserts (Eissa et al., 2015) – the ML model may underperform."

Finally I can say that problems such as the spatial (difference of point/station to grid/satellite) and the temporal (15 min satellite frequency vs 1 minute measurements integrated to hourly), seem to somehow dealt in a nice way with the ML training.

Minor

"Note that, since night-time is flagged as failing QC, 30% is a high requirement", I don't understand this maybe you could clarify.

We have rephrased this part; we hope it is clearer:

> "In the first setup, 100 test stations are chosen randomly from those passing QC for more than 30% of the hours over the test period (2018-07-01 to 2019-06-30). In other words, QC must be passed for at least 8 hours per day on average. As night-time is always flagged as failing QC, this is a stringent requirement."

---

## Author Comment (AC3)

comments:

The machine learning based techniques are very popular in recent years, and these methods do provide very good performance in various fields. And people start to question is it possilble for ML to replace the traditional physical method.

I think the authors tried to try to give some explanation from some extend. In this paper, the authors used physical method and ML based method to infer SSI from satellite images. And the authors gave an overall exploration of the MLP and analysed pitfalls and drawbacks of the method. It is very interesting that the result from MLP is better than CAMS from some aspects.

This paper is of high standard, I have some questions about some points, and hope the authors can consider.

1. There are many machine learning methods, and MLP is just a very basic ML method, why you chose this method? From the introduction part, I can't see some information about how MLP is used in the past researches, how it works in this field? Of course, this paper explained the performance of AI and physical method, but MLP can't represent AI techinque, could you explain why don't you use some other state-of-art AI techniques? I think some more information should be given in introduction part as well.

We did test some deeper architectures (2 and 3 hidden layers, some with more hidden neurons), but they didn't perform better. Following the principle of parsimony, we thus kept the simpler model (this is mentioned in section 3.3). Regarding other, more complex architectures, we were also guided by the principle of parsimony: if an MLP can do the job (and we consider it does since it already outperforms CAMS in training setup 1), there is no need to deploy a more complex model.

In addition, the dimension of the chosen input set did restrict a bit the list of adapted architectures: because the spatial extent of the input is only 3x3, using a CNN architecture would not make sense: the smallest size of a CNN kernel is usually 3x3. (Admittedly, we could have tested a RNN model to handle the 12 time steps in input.)

Most importantly, the goal of this work is to underline ML-specific validation issues, notably linked to generalization. More complex models (CNN, RNN, LSTM, Transformers, GAN, etc.) are trained the same way as MLP and therefore suffer from similar issues when it comes to generalization or out-of-domain performance. It is even likely that more complex models are more subject to overfitting and thus bad generalization (see for example *Generalizability of Neural Network-based Identification of PV in Aerial Images*, Ranalli 2023 for a related discussion).

We have explained this choice in the introduction:

> "Our objective is not to introduce a new retrieval method; hence, we have deliberately opted for a simple, fully connected architecture. This choice allows our conclusions regarding generalization to extend more effectively to the realm of complex networks (convolutional, recurrent, attention-based, generative, etc.), which are generally prone to encountering greater generalization challenges (Wang et al., 2017; Ranalli and Zech, 2023)"

We also agree that investigating more complex architecture is of interest. We have thus updated the perspective section (6.3):

> "The investigation of more sophisticated neural network architectures is also of interest and would become particularly relevant when dealing with larger input datasets. Architectures such as Convolutional Neural Networks (CNNs), Recurrent Neural Networks (RNNs), or Spatio-Temoral Transformers hold promise, especially when a broader context in both time and space is required. However, it is important to recognize that such complexity may raise the risk of generalization issues, as more complex models are generally more likely to overfit."

2. I think another index should also be considered, that is efficiency. What is the running time of MLP and CAMS respectively? This is also a very important index to see the performance of the two methods.

That is a good point. We've added a subsection about it: "3.4 running time" (see the updated manuscript

3. According to 5.3.3, as to ML methods, input training dataset plays very important part in the result, it should include all the necessary information it needs. So that is why a correlation analysis between input variable and target variable is necessary.

If we were doing in-depth feature engineering, we agree that a correlation analysis between potential predictors (or features) would be the first necessary step. Here, however, we opt to keep the ML model as simple as possible and to only use data from MSG (the other predictors are calendar-derived).

Just as the authors have analysis, in some locations, ML method shows poor performance because it doesn't have direct access to that information, is it because satellite can't cover that area which lead to this problem?

All the areas tested are well covered by the satellite, so it shouldn't explain bad performance. We rather think that this is due to these stations being out of the training domain.

---

## Author Comment (AC4)

This paper presents a study of the determination of surface solar irradiance from satellite using a multi-layer perceptron (MLP) compared to the state-of-the-art Copernicus Atmosphere Monitoring Service (CAMS) retrieval model. The paper is well written and well structured. The scientific question is relevant to the community and to the journal's field of application. However, the title does not accurately describe the content of the article, which deals with one particular model, namely an MLP with a hidden layer. Machine learning is a broad field that cannot be reduced to MLPs. In several places in the text, the conclusions are intended to be extended to machine learning, but they should be limited to the MLP used here.

The outcome of this work is to underline Machine Learning-specific validation issues primarily linked to generalization. These limitations are not specific to any type of ML model, we thus think that any model could be used to illustrate them.

Furthermore, more complex models are trained the same way as MLP and suffer from the same issues when it comes to generalization or out-of-domain performance. As a matter of fact, more complex models are more subject to overfitting and thus bad generalization; the conclusions obtained for MLP are therefore very likely applicable to more complex architectures (e.g. CNN, RNN, LSTM, Transformers, GAN).

We agree that this was not obvious in our introduction; we have updated the text to justify our choice better:

> "Our objective is not to introduce a new retrieval method; hence, we have deliberately opted for a simple, fully connected architecture. This choice allows our conclusions regarding generalization to extend more effectively to the realm of complex networks (convolutional, recurrent, attention-based, generative, etc.), which are generally prone to encountering greater generalization challenges (Wang et al., 2017; Ranalli and Zech, 2023)"

We also agree that investigating more complex architecture is of interest, and have thus updated the perspective section (6.3):

> "The investigation of more sophisticated neural network architectures is also of interest and would become particularly relevant when dealing with larger input datasets. Architectures such as Convolutional Neural Networks (CNNs), Recurrent Neural Networks (RNNs), or Spatio-Temoral Transformers hold promise, especially when a broader context in both time and space is required. However, it is important to recognize that such complexity may raise the risk of generalization issues, as more complex models are generally more likely to overfit."

The sensitivity study is interesting, but raises some questions about the selection of predictors for the MLP model. For example, it is noted that not including AOD at 500nm leads to underestimation. So why not include AOD 500 as a predictor? In general, the comparison between MLP and CAMS seems biased, since CAMS seems to have access to more variables (especially thanks to the clear-sky model). For a neutral comparison, wouldn't it be better to list all the variables used by CAMS (including those hidden in the clear-sky model) and use them as predictors of the MLP model? How can MLP be expected to take into account the effect of AOD if it has no access to AOD values? The same applies to albedo.

Adding AOD 500 and albedo as predictors would indeed probably improve the performance of the model. Similarly, we could imitate CAMS's structure, and decompose the problem in the computation of a cloud index and of a clear sky (ie learn a clear sky index instead of directly the GHI). It would also be interesting to see if the normalization of MSG radiance into reflectance could help the model learn. Generally, there is a lot of feature engineering that could be done to make the model stronger. This is, however, not the purpose of this study, and because it would be a sizeable work, we believe it should be kept for future work.

That being said, we agree that a possible approach would have been to use exactly the same data as CAMS, i.e. adding AOD and albedo as input; this data, however, comes from complex numerical models (MODIS, Copernicus). In this study, we preferred to keep the ML model as simple as possible and to only use data from MSG (the other predictors are calendar-derived).

Regarding the comparison with CAMS, it is true that the two models use slightly different data. CAMS has some knowledge of the albedo and AOD while the ML model has access to more MSG pixels (even though strictly speaking, CAMS module McCloud does use neighboring pixels to determine albedo). CAMS should therefore only be seen as a reference that allows us to relativize the ML model performance in each station.

We agree that this approach may impact the results discussed in section 5.2.2 (impact of aerosols). We added a discussion to that section:

> "This result is somewhat expected, as CAMS model integrates some information about the AOD (through McClear), whereas ML model does not. Adding AOD-related predictors to the neural network may help decrease the performance gap between the two methods for clear skies."

The ML model is assigned 9 pixels, whereas for CAMS, only 1 pixel is used. Why not considering an average of the 9 pixels for CAMS? This would give an idea of the contribution of MLP compared with a simple spatial average.

We agree that some simple post-processing of CAMS – such as smoothing as you suggest – is likely to decrease its RMSE. We preferred, however, to use CAMS 'as is' because that is how it is evaluated in most studies and, arguably, used by the majority of users. In addition, using *raw* CAMS output allows us to compare to orthogonal approaches: one purely ML-based and one without any statistical processing.

Here are some detailed comments:

- l. 52: Why did the authors only use 3 bands out of the 12 available? And why not use the HRV channel which is always available for France?

  Testing the impact of other channels on a ML model performance certainly makes sense. The ability to fusion several channels is even one motivation for using machine learning. For this work, however, we only had access to a large database of past data for these three channels.

  In addition, these three channels are the ones used by heliosat2 (it uses 0.6 and 0.8) and heliosat4 (it uses all three, and, in some rare cases, 3.7um as well), so we can expect them to be the most important.

  We have added a note to section 3.2 to mention that using other bands would be of interest:

  "These are the channels mainly used by Heliosat 2 and Heliosat 4. Other wavelengths may nonetheless be useful to a machine-learning-based model, and their impact on model performance should be explored in future work."

- l. 60: The link at the bottom of the page that describes the instrument is broken.

  Noted, thanks. I think it is fixed – at least it works for me. I will double-check once we get an online manuscript.

- l. 100: « The inverse transformation is applied to the network predictions before starting to analyze its performance ». As it is not explicit can the authors confirm that they apply the inverse transformation with the mean irradiance computed from the training set? And why do they normalize by the mean instead of removing the mean and normalizing with the standard deviation?

  We indeed use the average computed for the training. We have specified it in the text:

  "The inverse transformation (also with the average irradiance over the training period) is applied to the network predictions."

- About the MLP structure: How the configuration (number of neurons, activations, initialisation) was chosen?

  We choose default/standard approaches for Initialization and activation. For the number of hidden layers and their size (number of hidden neurons), we tested a few configurations (64, 64x64, and 64x64x64). All had similar validation performances, so we chose the simplest one.

- l. 134: « Regularization is implemented through an early stopping procedure, which stop training if the validation error does not decrease for more than 20 epochs. ». The description of the regularization is incomplete. What is the minimum variation used to stop the training?

  It is necessary to define a minimum variation if early stopping is done with the training set. Here we use a validation set, there is, therefore, no minimum variation: the training stops if the validation error has not strictly decreased for 20 epochs.

- l. 136: « Because the last layer uses a linear activation function,… ». Why choosing a linear activation while a RELU should resolve the positiveness problem?

  It is generally recommended to use a linear activation function as the last layer for regression problems. We agree that reLu would solve the negative prediction issue, but we were worried that it may perturbate the gradient descent and cause a higher bias. Because using a linear activation function only leads to very few negative values, we chose to stick to the 'default'.

- l. 179: Are 8 years of data really needed for the training? What about the sensitivity about the size of the training set?

It is a very good point, thanks. We have added a training setup where we decrease the number of training years while keeping the number of training stations equal to 129 (the maximum). The results are discussed in section 5.3.1 "Impact of the number of training years" (see the updated manuscript).

- Table 3: The MBE values seem to be incorrect. They are not coherent with Fig. 4b

  Indeed we inverted MBE and SDE. Thank you for spotting this.

- l. 235: CAM instead of CAMS

  Indeed, thanks.

- l. 246: « Furthermore, CAMS seems to handle situations for which the clear-sky is close to one better than ML model». Is that seen from the yellow "spot" near kc=1 that is more diffuse for ML? This statement is not clear.

  That is indeed what we meant, but looking closely at the graph, it is not that clear and we may have been influenced by the results of 5.2 in this analysis. We have removed the sentence.

- Figure 3: « ML model (a and c) and for CAMS (b and d) » should be « CAMS model (a and c) and for ML (b and d) »

  Indeed, thanks.

- l. 256: Contradiction with statement line 233 while MBE was of the order of 50W.M-2

  There was a mistake in table 3 – now should be coherent.

- l. 263: « Figure 4 ». It should be Fig. 5.

  Indeed, thanks.

- l. 315: « As discussed in Section 4.2.2, we repeat the experiment 20 times for each choice of $N$, with different randomly chosen training stations at each iteration ». Is the subsampling of stations ensuring that they are well distributed in space? How do the authors achieve this?

  The subsampling is purely random and therefore there is no guarantee that the stations are well distributed in space. Training setup 3 (previously setup 2) therefore somehow overlaps with training setup 4 (previously setup 3).

  Now that we added training setup 2 (with decreasing number of training year), this further suggests that indeed the bad performance for some choice of 20 stations in training setup 3 comes from the stations' location more than the size of the training set.

  We mention this in the results section 5.3.2:

  "Put in perspective with the results of Section 5.3.1, this suggests that the issue is not the size of the training set, but the location of the training stations."

---

## Author Response (AR1)

Dear editor, dear reviewers,

Thank you very much for your feedback. We greatly appreciate the time and effort you committed to reviewing this manuscript. We have endeavored to answer all your points and provided a detailed answer to each of them. We have provided our replies as responses to each reviewer in supplement pdfs.

We made several changes to the original text, as can be seen in the annotated manuscript. The major changes are as follows:
- We added a subsection to section 3 to discuss the running time of the ML model.
- We added a new training setup to analyze the impact of the number of training years on the performance of the ML model. Method and results sections were updated to incorporate this setup.
- We insisted in the introduction, methods, and conclusion on the reason why we chose a MLP and why we think it can be representative of the risk of using ML in general – even for more complex neural networks.
-

Multiple *minor* updates were also made to the text following the remarks of the reviewers. For an explained list of each change, please refer to our responses to each reviewer.